# Toward Sustainability of the Aqueous Phase Reforming of Wastewater: Heat Recovery and Integration

**Francisco Heras** [1,*] , **Adriana S. de Oliveira** [1,2], **José A. Baeza** [1] , **Luisa Calvo** [1] , **Víctor R. Ferro** [1]
**and Miguel A. Gilarranz** [1]

1   Department of Chemical Engineering, Universidad Autónoma de Madrid,
    Ciudad Universitaria de Cantoblanco, 28049 Madrid, Spain
2   Instituto IMDEA Energía, Avda. Ramón de la Sagra, 3, Parque Tecnológico de Móstoles, 28935 Móstoles, Spain
*   Correspondence: fran.heras@uam.es

**Featured Application: Aqueous-phase reforming is a promising process for hydrogen production; however, the energy demand is one of the main drawbacks of its potential feasibility. The mass and heat balances shown in this work suggest that it could be possible to achieve the heat self-covering of the process as long as adequate operating conditions are used.**

**Abstract:** Aqueous-phase reforming has been revealed as a novel, interesting and efficient process for the treatment of wastewater containing organic pollutants. However, due to the relatively severe operating conditions (above 15 bar and 200 °C), this process could become economically competitive if any solution for energy or material valorization is implemented. Most research has been devoted to direct the process to $H_2$ production as an alternative to reach economic sustainability, but the results obtained were not competitive in the current market of hydrogen and syngas. In this work, a preliminary simulation study (using Aspen HYSYS software) of the process heat balance in different conditions was implemented to induce a heat integration that would allow the auto-sustainability of the process, even generating in some cases an excess of energy that could constitute an opportunity for a positive economic balance. The results showed that this approach would only be possible by maximizing the methane production to the detriment of hydrogen production.

**Keywords:** aqueous-phase reforming; process sustainability; heat integration; hydrogen

## 1. Introduction

Aqueous-phase reforming (APR) is a catalytic process for the transformation of oxygenated organic compounds into hydrogen; it also generates alkanes as a byproduct [1,2]. In general terms, the process is relatively similar to steam reforming, but is carried out in the liquid water phase at temperatures around 200 °C [3,4].

In the context of hydrogen economy, APR has been studied as a method to produce renewable hydrogen [5]. Many different feedstocks have been studied [6], including sugar alcohols (such as glycerol, xylitol or sorbitol, among others) [7], bio-oils, bio-refinery residues [8] and biomasses (including waste biomass) [3,7–11], the last of which has recently been given special attention. The best performance of APR, in terms of $H_2$ production, was reported to be obtained using organic feedstock compounds with a C:O ratio near 1:1 [4,12], but the results in terms of conversion and selectivity to hydrogen were highly dependent on the type of catalyst and operating conditions. For instance, solutions of 10%w xylitol have been used as feedstocks for APR in similar operating conditions (220–225 °C, 29 bar), yielding different results depending on the catalysts used. Murzin et al. [13] reported 70–90% $H_2$ selectivity using a 2.5%w Pt/carbon commercial catalyst while Kirilin et al. [14] reached 65–75% using homemade 5% Pt/Al$_2$O$_3$ catalysts and Kim et al. [15] obtained values around 75% with a variety of homemade 7% Pt/carbon catalysts. In most of cases, feed-to-catalyst ratios of 50–150 mL/g [16,17] for batch operation and 0.1–6 h$^{-1}$ for continuous

operation [14,18,19] were commonly used. The hydrogen content in an APR gas product is highly conditioned by operating conditions (temperature, pressure and feed concentration), influencing not only $H_2$ but also CO, $CO_2$ and $CH_4$ formation, as was revealed by the thermodynamic analysis of glycerol APR [20]. Likewise, the concentration of substrate in the APR feed stream was shown to have an important influence. In general terms, aqueous phase reforming needs a diluted feed stream; however, a moderate concentration increase has been found to lead to higher substrate conversion at the expense of lower hydrogen yield [1,21,22]. In this context, APR has also been applied to the treatment of low-concentration wastewater bearing biomass-derived pollutants, which made it possible to decrease wastewater organic load and generate a valuable gas stream from waste [8,23].

Most works on APR in the literature are based on experimental work and are focused on catalyst development and the influence of feedstock and reaction conditions in organic matter conversion and gas product yield. Several studies can also be found in the literature about process modeling and simulation, with the aim of assessing APR process feasibility from a technical and economic perspective. These works show consensus of the fact that temperature and heat requirements for the APR process require compensation to make the process economically feasible. The most common option has been the use of $H_2$-rich gas generated for commodity applications; however, the heat integration and/or the heat valorization of alkane byproducts produced is also an interesting option [13]. Likewise, the price of hydrogen does not compensate sufficiently for the cost of some marketable feedstocks proposed as substrates in APR [24]. One of the challenges in the application of APR to the treatment of wastewater is that the low organic carbon, in comparison to other proposed feedstocks, limits the amount of gas that can be produced and used to achieve auto-sustainability of heat demand of the process. This aspect has also been studied for other wastewater treatment processes, enabling the production of flue gas but using other mature technologies, such as steam reforming (SR). Thus, Lee et al. [25] studied the integration of heat and power generation in a wastewater treatment plant on the basis of biogas valorization, concluding that ca. 47 and 100% of power and heat demands could be covered, respectively. On the other hand, the production of 80 kg/h hydrogen could be achieved by APR at 250 °C/50 bar with lower costs than by SR at 550 °C/1 bar, using glycerol as a feedstock in both cases [26].

This study deals with the evaluation of heat integration and the sustainability of APR applied to the treatment of wastewater. Heat balances were calculated under different operation scenarios in terms of reactor operating pressure, conversion of reforming and methanation reactions and organic matter concentration in feedstock to determine main contributions to heat demand and production. Sensitivity analyses were carried out for conversion in reforming and methanation reactions as well as operating pressure. All results were obtained via simulation using Aspen HYSYS V.10® software.

## 2. Materials and Methods

Previous experimental works [14,24] showed that the concentration of organic matter in the wastewater subjected to APR must be maintained below 5% in order to maximize substrate conversion and hydrogen production. On the other hand, low-concentration feedstocks showed the disadvantage of increased heat demand in the reactor, mainly due to preheating needs and higher water evaporation. In addition, organic matter concentration below 5% could be considered representative of wastewater streams generated in the food and beverage industries. Therefore, the case study considered for the simulated APR process was the treatment of a 180 $m^3$/h stream containing 1–2%w of carbohydrate-like organic matter.

The concentration of the wastewater was considered as a process variable. Sorbitol was selected in Aspen HYSYS databases as a substrate because it has been widely studied as an organic matter model compound in several works on APR [14,24,27], facilitating the comparison of results.

The reactions involved in APR are mainly dehydrogenation and the breakage of C-C bonds of organic compounds to produce hydrogen and carbon monoxide, water–gas shift (WGS), methanation of $CO/CO_2$ with $H_2$ and other minor reactions such as Fischer–Tropsch, dehydration and hydrogenation [24,28,29]. Table 1 shows the main reactions involved in the APR process.

**Table 1.** Main reactions involved in the APR process [23,27,28].

| Reforming | $C_nH_{2n+2}O_n + nH_2O \Leftrightarrow nCO + (2n + 1)H_2$ |
| --- | --- |
| **Water–Gas Shift** | $CO + H_2O \Leftrightarrow CO_2 + H_2$ |
| **Methanation** | $CO_2 + 4H_2 \Leftrightarrow CH_4 + 2H_2O$<br>$CO + 3H_2 \Leftrightarrow CH_4 + H_2O$ |
| **Fischer–Tropsch Hydrogenation Dehydration** | $(2n + 1)H_2 + nCO \rightarrow C_nH_{2n+2} + nH_2O$<br>$2nH_2 + nCO \rightarrow C_nH_{2n} + nH_2O$<br>$C_nH_{2n+2}O_n + nH_2 \Leftrightarrow C_nH_{2n} + H_2O$ |

Attending to both reaction enthalpy and the contribution to the global reaction scheme, two reactions were taken into consideration for the purpose of this work: reforming and methanation, shown in Equations (1) and (2), respectively, for the case of sorbitol. In addition, these two reactions represent a route for the transformation of substrate into gas products.

$$C_6H_{14}O_6 + 6 \cdot H_2O \leftrightarrow 6 \cdot CO_2 + 13 \cdot H_2 \qquad \Delta H° = 229.9 \text{ kJmol}^{-1} \qquad (1)$$

$$CO_2 + 4 \cdot H_2 \leftrightarrow CH_4 + 6 \cdot H_2O \qquad \Delta H° = -163.0 \text{ kJmol}^{-1} \qquad (2)$$

As expressed in Equation (1), the so-called reforming reaction includes both steps of the whole reaction: decomposition of sorbitol to $H_2$ and CO and further oxidation of CO to $CO_2$ through water–gas shift. For that reason, WGS was not considered as an independent reaction in the study.

Aspen HYSYS V12 was used as a calculation tool. For thermodynamic property estimation, the Peng–Robinson–Stryjek–Vera (PRSV) model was selected to reflect the non-ideal behavior of the vapor/gas phase. This equation of state has previously shown good results in the property estimation of the compounds involved in APR [20,23]. Figure 1 shows a block flow diagram of the proposed APR process, in which heat consumption and possible heat production/recovery steps are indicated. This process scheme allowed a low-limit analysis (i.e., most limiting situation) for the case where the gas stream was valorized through combustion, with no other value recovery.

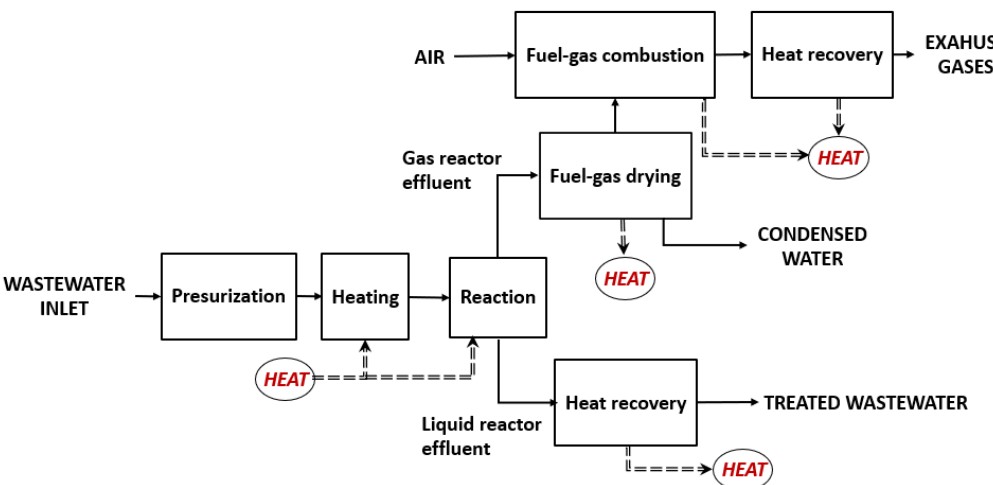

**Figure 1.** APR process block diagram.

The simulated process comprised the following steps:

- Feed conditioning: temperature and pressure of wastewater was increased to usual operating conditions in APR: 220 °C and 30 bar in the current work [3,4,23,30].
- Reactor thermal operation mode: the reactions involved in the APR process were considered to take place in an isothermal and isobaric regime, generating the gas-product stream and treated wastewater.
- Gas drying: water from the gas-product stream leaving the reactor was removed by condensation at high pressure in order to maximize heat recovery (global heat transfer coefficient, U, increases with temperature), as has also been proposed in previous studies [2,13].
- Heat recovery from treated wastewater: treated wastewater left the reactor at reaction temperature, and heat was recovered from this stream by exchange.
- Combustion of gas-product stream and heat recovery: the gas-product stream containing $H_2$, $CH_4$ and $CO_2$ was considered a fuel gas that was burned in a fired heater for heat recovery.

The ability of the process to cover heat intake was assessed through the parameter of heat covering (HC), which has been defined for the purpose of this work as follows:

$$HC\ (\%) = 100 \cdot (\Sigma Qi / \Sigma Qc) \tag{3}$$

where Qi is the heat flow in the steps in Figure 1 where heat is recovered, and Qc is the heat flow in the steps where heat is consumed. For the heat-flow expression, a conventional criterion has been used, i.e., Q < 0 for recovery and Q > 0 for consumption. Thus, the HC parameter took positive values when the process showed a net heat deficit and took negative values when the total heat generated was higher than the total heat consumption.

The most significant heat consumptions in the process were:

- Pressurized inlet wastewater preheating to reach the reactor operating temperature: in the industrial process, it is usual to cover it (at least partially) with heat interchange with the reactor outlet stream.
- Reactor heat demand (reactor duty): the most important heat consumption. The significance of this contribution did not come from the reaction enthalpy (which is not high) but from water evaporation, which is a key aspect for cost-efficient operation when processing diluted feeds [23].

On the other hand, heat could be recovered from:

- Treated reactor effluent: can be used for reactor-feed-stream preheating as indicated above.
- Condensation of water present in gas stream from the reactor; it was highly dependent on the amount of water evaporated in the reactor.
- Gas stream from the APR reactor: by combustion of this stream, a large amount of heat could be recovered both in the furnace unit and from hot exhaust gases.

### 2.1. Process Heat Balance under Different APR Operating Conditions

The objective of this section was to perform a sensitive analysis to evaluate the influence of the concentration of organic matter in wastewater and conversion of the reforming and methanation reactions on the overall process heat balance. The conversion of each reaction was managed as an independent variable in order to generate different possible operation scenarios, allowing determination of the interval with the most favorable heat balance. Figure 2 shows the Aspen HYSYS simulation diagram used for heat-balance analysis.

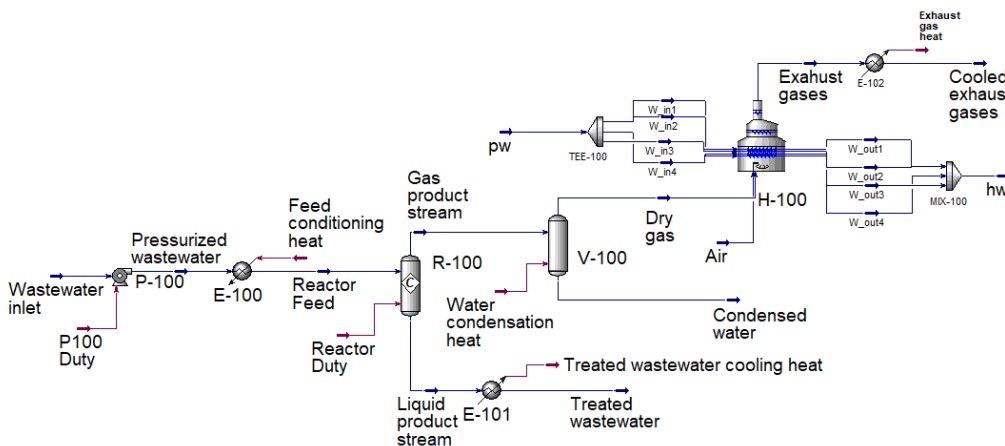

**Figure 2.** Aspen HYSYS process flow diagram used for heat-balance analysis.

On the other hand, the conversion values in the reactions involved influenced the concentration of $H_2$ and alkanes, mainly $CH_4$, in the produced gas, leading to different heating values of the stream. Reforming conversion represents the substrate conversion achieved, which has been found to reach values well above 70% in a number of case studies [24,31,32]; therefore, a range from 70 to 100% was considered for the sensitive analysis. Methanation reaction conversion was studied from 0 to 55%, with the latter corresponding approximately to the equilibrium conversion at selected operating conditions.

A conversion reactor model was selected to study the influence in the heat balance of conversion in reforming and methanation reactions. A pressurized water stream (pw) was included in the flow diagram to solve the fired heater unit and obtain information about the heat flow that could be recovered in this unit. The operating conditions for the simulation and studied variables are compiled in Table 2.

**Table 2.** Operating conditions and studied variables for process heat balance analysis.

| **Operating conditions** | |
|---|---:|
| Wastewater inlet flowrate (m$^3$/h) | 180 |
| Wastewater inlet temperature (°C) | 20 |
| Wastewater inlet pressure (bar) | 1 |
| APR reactor temperature (°C) | 220 |
| APR reactor pressure (bar) | 30 |
| Treated wastewater temperature (°C) | 25 |
| Exhaust gases temperature (°C) | 110 |
| Fired heater parameters (%) | |
|     Combustion efficiency | 65 |
|     Oxygen excess | 10 |
| **Studied variables (independent)** | |
| Wastewater inlet concentration (%w organic matter) | 1–2 |
| Reaction conversion (%) | |
|     Reforming | 70–100 |
|     Methanation | 0–55 |

*2.2. Heat Integration Study*

After the heat balance analysis, operating conditions leading to better HC results were selected for a heat integration study in order to evaluate if it is possible to achieve heat covering thanks to the energy in the streams leaving the APR reactor. Among the many heat integration strategies that could be proposed, a basic alternative that allows achievement of the aforementioned target was used, taking into account that other, more complex strategies (including Pinch Analysis and other methodologies) could reach the optimal heat recovery.

The Aspen HYSYS simulation diagram for the heat integration strategy used is shown in Figure 3. In contrast to heat balance analysis, heat exchange models were used for heating and cooling steps and the fired heater was used for the reactor-feed thermal conditioning. The liquid stream from the APR was used in reactor-feed preheating while the gas stream, after drying, was divided in two new streams: the first was used as fuel in H-100 to produce the necessary heat to cover the reactor duty (not integrated in PFD but considered in heat balance), and the rest (recovered fuel gas) was considered a fuel gas excess that could be used for other purposes. The heat from water condensation and exhaust gases was not easy to recover from a technical point of view for the APR process, but could be recovered to cover other plant demands.

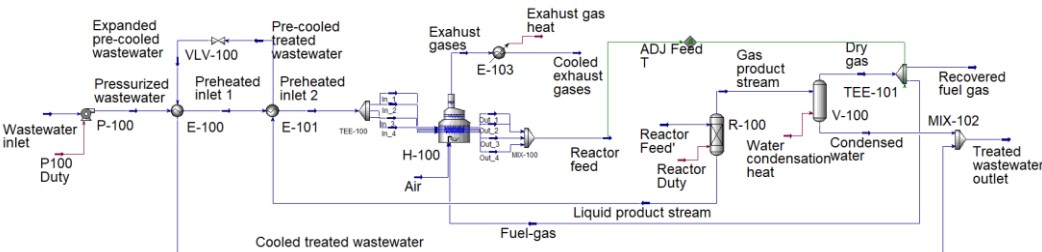

**Figure 3.** Aspen HYSYS process flow diagram used for heat integration.

## 3. Results and Discussion

### 3.1. Process Heat Balance at Different APR Operating Conditions

Figure 4 shows the flow and composition (dry basis) of the gas stream produced for different values of reforming and methanation reaction conversion for wastewater containing 1% of organic matter. At any reforming conversion value, the lowest $H_2$ production was obtained for higher extension of methanation reaction due to hydrogen consumption in this reaction. On the contrary, the increase in methanation conversion caused an increase in $CO_2$ concentration in the gas stream: in this case, much slighter than that observed for $H_2$ concentration. The differences in the stoichiometry of reforming and methanation reactions resulted in lower gas flow at low reforming conversion and high methanation conversion, even though the stream composition was not altered.

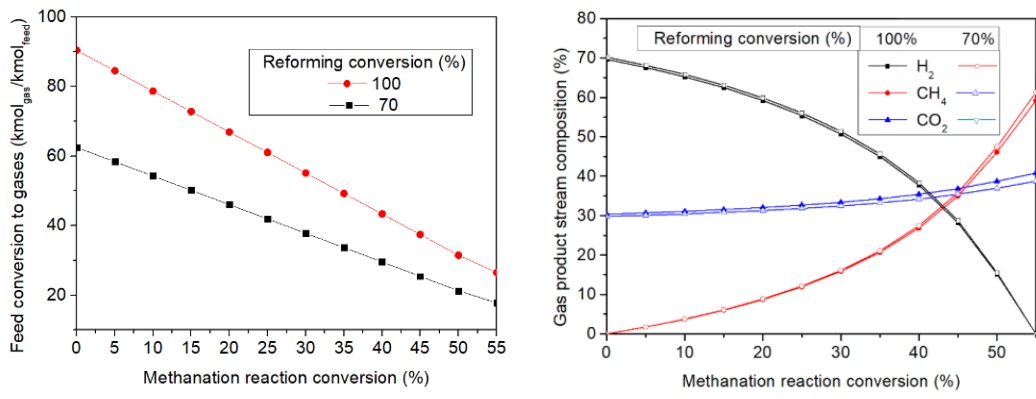

**Figure 4.** Gas production and composition (dry basis) at the APR reactor.

As explained previously, the evaporation of water in the reactor played an important role in overall heat balance. The molar flow of evaporated water in the reactor vs. reforming and methanation conversion is plotted in Figure 5. Due to stoichiometry of the reactions involved, at constant pressure, the gas-stream molar flow was higher for high reforming conversion. Higher total gas flow resulted in higher water flow because the water molar fraction in this stream was conditioned by equilibrium.

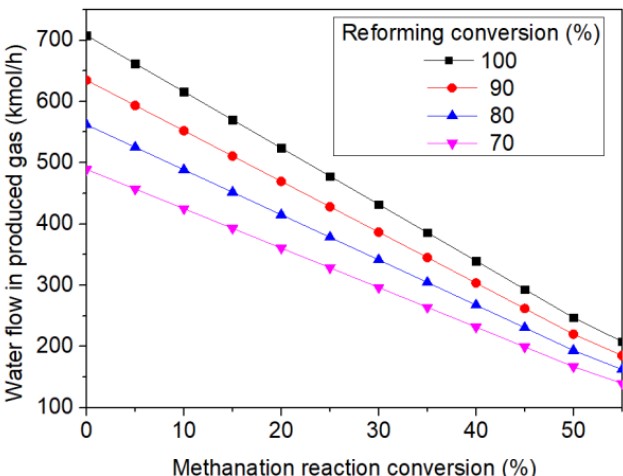

**Figure 5.** Molar flow of water evaporated in APR reactor vs. reforming and methanation conversion.

As the evaporation of water is the most important contribution to energy demand in the reactor, the influence of operating pressure in the flow of evaporated water is relevant. Evaporated water vs. reactor operating pressure values for different reforming reaction conversion are plotted in Figure 6. At every pressure and reforming conversion, a mean value of evaporated water with methanation conversion in the interval between 0 and 55% was considered. A decrease in water molar flow could be observed when pressure was increased up to 50 bar. A moderate increase of pressure from 30 to 35 bar led to a 50% reduction in water flow. From 45 bar, a minor effect of pressure could be observed. However, an operating pressure of 50 bar was selected to minimize energy demand from the evaporation of water, since the compression of wastewater was a minor contribution to overall energy demand. These results are in concordance with those reported by Sladkovskiy et al. [23]; the evaluation of APR of sorbitol by simulation with Aspen Plus in the 30 to 50 bar range suggested that the reactor operating pressure must be maintained over 40 bar to minimize the heat reactor duty.

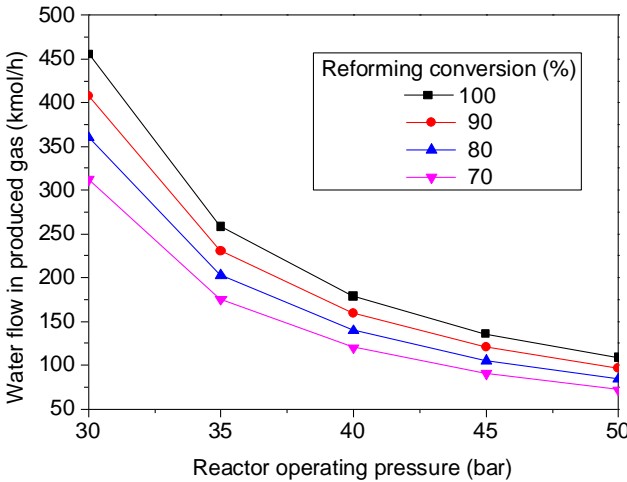

**Figure 6.** Molar flow of water evaporated in APR reactor vs. operating pressure.

Figure 7 shows HC for the APR process and its dependence on the conversion of reforming and methanation reactions. The negative values obtained for the HC parameter ($-6.7$ to $-13.5$%) indicated a net heat excess within the whole range considered for reforming and methanation reaction conversion. A higher excess was obtained at a higher extension of reforming reaction due to a larger production of gas. Net heat excess was also higher for high methanation conversion due to the higher heating value of methane in

comparison to hydrogen. The steepness of the contour map in the y-axes indicated that methanation has a higher impact in heat excess than reforming. These results indicated that the orientation of the APR process, i.e., towards hydrogen or towards alkanes, has to take into consideration whether the produced gas will be sold as a commodity or valorized in the APR plant. In the latter case, higher HC is achieved by promoting alkane production.

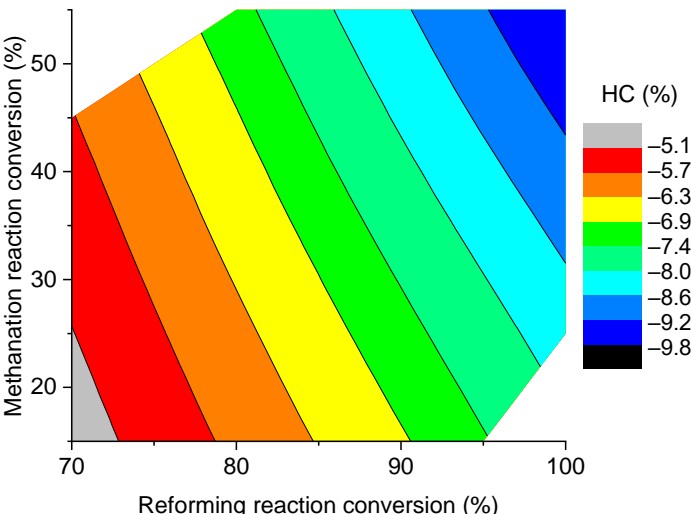

**Figure 7.** HC for different reforming and methanation conversion values.

Finally, the influence of wastewater concentration in the heat balance was studied using different reactor operating pressures in the 30–50 bar interval (every 5 bar). Table 3 shows the maximum HC achievable, in the whole pressure interval studied and for an organic matter concentration in the feedstock of 1 and 2%, for total conversion at reforming reaction and extreme values of methanation reaction conversion. Due to the higher gas production, higher values of HC were obtained when 2% inlet concentration was considered, reaching HC values well above 20%. After HC vs. pressure sensitivity studies, while best results in heat balance were obtained operating at 35 bar for 1% inlet concentration, the highest result was reached at 45 bar when the concentration was increased to 2%. In any case, since the compression of wastewater is not a high-cost operation, higher operating pressure could be achieved and would be largely compensated by the increase of HC. The values of operating conditions found as optimal are in agreement with those used in most reported works on the APR of sorbitol and similar polyols (xylitol, etc.), i.e., 30–50 bar pressure range and organic matter concentration below 10% [14,23,27]. Likewise, the results are concordant with those reported for the simulation of the APR of 10% sorbitol [23]. The authors found similar results in terms of process heat balance but concluded that the main limitation for process feasibility was the cost of feedstock, amounting to around 98% of total operating cost. The obtained HC results indicated that the application could be extended to a wider range of operating conditions. In this sense, when no cost was considered for the feedstock and an adequate strategy was used for heat recovery, it was possible to achieve the process of self-covering and use the net heat excess to decrease operating costs of steam generation or other usual alternatives in process engineering, as long as the operating conditions are above certain values depending on the feedstock composition.

**Table 3.** Maximum HC values for different concentrations of wastewater inlet.

| Inlet Concentration (%w) | P (Bar) | Conversion (%) Reforming/Methanation | HC (%) |
|---|---|---|---|
| 1 | 40 | 100/0 | −7.2 |
|   | 35 | 100/54 | −9.9 |
| 2 | 50 | 100/0 | −22.2 |
|   | 45 | 100/54 | −29.5 |

*3.2. Heat Integration Study*

Based on the results obtained in the heat-balance analysis at different APR conditions, heat integration was proposed for four different process operation scenarios, aiming at checking the potential heat self-covering of the process. The main target was to use the heat contained in the streams, leaving the APR reactor to cover thermal conditioning of the reactor feed because it is the main source of energy consumption. Table 4 summarizes the operating conditions for each process scenario studied.

**Table 4.** Operating conditions for different scenarios considered in heat integration analysis.

|  | Scenario | | | |
|---|---|---|---|---|
|  | 1 | 2 | 3 | 4 |
| Wastewater concentration (%) | 1 | 1 | 2 | 2 |
| APR reactor pressure (bar) | 40 | 35 | 50 | 45 |
| Reforming conversion (%) | 100 | 100 | 100 | 100 |
| Methanation conversion (%) | 0 | 54 | 0 | 54 |

Table 5 shows the results of heat integration for the different scenarios proposed. Since the thermal conditioning of reactor feed stream was fully covered by proposed heat integration, the reactor duty (energy to supply to the reactor to maintain the operating temperature) became the highest heat demand of the process. With the model and conditions used, energy consumption by endothermic reforming reaction was higher than energy consumption by water vaporization. In the case of scenarios 1 and 3, despite the high operating pressure, the reactor duty was higher than in scenarios 2 and 4, in which methane was formed.

Considering that the recovered fuel gas could be used in a boiler to produce steam for reactor heating (not included in the simulation presented in this work) with efficiency values of 80 and 70% (combustion and steam generation, respectively) [32], it was possible to calculate the proportion of reactor duty that could be supplied by the recovered fuel gas; the results are shown in Table 5 ("Energy recovery"). With this proposal, scenarios 2 and 4 yielded a net energy excess (recovered fuel gas excess) after reactor-duty covering that could be used to produce energy for other uses.

In scenario 1 (0% methanation and 1% feed concentration), the reactor duty could not be covered through the recovered fuel gas stream, whereas in the other scenarios, this stream contained enough energy for reactor-duty covering and there was excess that could be used for other process needs. For the scenarios with high methane production (2 and 4), high values of energy excess were observed. On the contrary, when methanation reaction was not favored (scenarios 1 and 3), the high concentration of $CO_2$ in recovered fuel gas led to a lower heating value. Therefore, the promotion of methanation reactions was needed to achieve energy sustainability of the treatment of diluted wastewater by APR.

**Table 5.** Heat integration results for the proposed operating scenarios.

| | Scenario 1 | Scenario 2 | Scenario 3 | Scenario 4 |
|---|---|---|---|---|
| Exhaust gases T (°C) | 514.4 | 567.3 | 510.5 | 558.5 |
| **Main gas streams flows (kmol/h)** | | | | |
| Exhaust gases | | | | |
| Fuel gas | 72.99 | 17.23 | 92.59 | 18.50 |
| Recovered fuel gas | 102.80 | 33.12 | 267.10 | 84.39 |
| **Main heat flows (kW)** | | | | |
| Reactor duty | 4566.0 | 773.9 | 7277.0 | 582.5 |
| Water condensation heat | −4160.0 | −1675.0 | −5673.0 | −2005.0 |
| Exhaust gas heat | −700.4 | −556.0 | −875.6 | −580.2 |
| **Main heat flows (kWh/m$^3$ of treated wastewater)** | | | | |
| Reactor duty | 25.5 | 4.3 | 40.7 | 3.3 |
| Water condensation heat | −23.2 | −9.4 | −31.7 | −11.2 |
| Exhaust gas heat | −3.9 | −3.1 | −4.9 | −3.2 |
| **Energy recovery** | | | | |
| Reactor duty (kW) | 4566.0 | 773.9 | 7277.0 | 582.5 |
| Reactor duty supplied by recovered fuel gas energy (%) | 70.7 | 100 | 100 | 100 |
| Net heat excess after reactor-duty covering (kW) | 0 | 2099.3 | 1037.7 | 6696.7 |

## 4. Conclusions

The main energy demand of the whole APR process came from the reactor and was strongly dependent upon the water vaporization. To reduce this energy demand, it is indispensable to select an adequate operating pressure and other variables to promote methanation reactions. Additionally, the maximization of methane production allowed process heat sustainability to be reached. In this case, the process heat balances indicated that enough energy was available for both process self-covering and the generation of a net excess of 7–30%. Feedstock concentration showed a significant influence, with a limit value of 1% for achieving energy self-covering. Optimal operation conditions were also influenced by feedstock concentration, with a required reaction pressure of 35–40 bar for 1% and 45–50 bar for 2%. The results suggested that the APR process could achieve a competitive position versus other water-treatment processes with a different approach to what has been usual in previous works based on $H_2$ production.

**Author Contributions:** Conceptualization, F.H. and V.R.F.; data acquisition and curation, F.H.; formal analysis, F.H. and V.R.F.; validation, A.S.d.O., J.A.B., L.C. and V.R.F.; supervision, F.H. and M.A.G.; writing—original draft, F.H., A.S.d.O. and J.A.B.; writing—review and editing, F.H., L.C., V.R.F. and M.A.G. All authors have read and agreed to the published version of the manuscript.

**Funding:** This research received no external funding.

**Institutional Review Board Statement:** Not applicable.

**Informed Consent Statement:** Not applicable.

**Conflicts of Interest:** The authors declare no conflict of interest.

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
