# Peer review of "Toward Sustainability of the Aqueous Phase Reforming of Wastewater: Heat Recovery and Integration"

_applsci, doi:10.3390/app122010424_

Round 1

Reviewer 1 Report

Reviewer’s comments on manuscript applsci 1946764

Manuscript has a clear structure and is written in good English. A few formatting issues were found, listed in the Comments and Suggestions below. Aim of the study is interesting, fits well to Applied Science scope but from relevance and contribution in the scientific field the study needs substantial enrichment by:

-          more detailed and focused literature survey proving the study aims are novel and might truly present valuable contribution in the field.

-          Calculation scope and complexity must be enriched. Simple balance calculations, however their results seem interesting, should be supplemented with detailed parametric studies or optimization.

-          A wider discussion, encompassing both own results and those presented in relevant related studies should follow the results – see line 83 where comparison of results is promised but this promise is not kept. Comparison of results with other studies is omitted, which is one of the main drawbacks of the manuscript.

At present the study resembles more an extended computation exercise report than a scientific manuscript. Please do remember that Applied Sciences is a well-established scientific journal in Q2 JCR, publishing relevant research. Therefore, a thorough manuscript revision is needed.

Comments and suggestions regarding formatting:

Please unify the formatting (“10%” or “10 %” as an example). Please do use correct symbols (line 42 and 121 should be ”°C” as an example). Please refer to the equation numbers in the accompanying text.

Table 4: see “Methanation conversión”

Author Response

The authors are grateful for the comments and interest that the reviewer has paid to the manuscript.

Manuscript has a clear structure and is written in good English. A few formatting issues were found, listed in the Comments and Suggestions below. Aim of the study is interesting, fits well to Applied Science scope but from relevance and contribution in the scientific field the study needs substantial enrichment by:

more detailed and focused literature survey proving the study aims are novel and might truly present valuable contribution in the field.

According to the reviewer´s comment, literature cited has been updated. Some recent references have been incorporated and discussed, mainly to compare the results of the current work, as it can be seen in the annotated manuscript. The new references are:

  • Rossetti, A. Tripodi, Catalytic Production of Renewable Hydrogen for Use in Fuel Cells: A Review Study. Topics Catal. 2022. DOI: doi.org/10.1007/s11244-022-01563-z
  • Zoppi, G. Pipitone, R. Pirone, S. Bensaid, Aqueous phase reforming process for the valorization of wastewater streams: Application to different industrial scenarios. Catal. Today 2022, 387, 224-236; DOI: doi.org/10.1016/j.cattod.2021.06.002
  • Khodabandehloo, A. Larimi, F. Khorasheh, Comparative process modeling and techno-economic evaluation of renewable hydrogen production by glycerol reforming in aqueous and gaseous phases. Energy Conv. Manag. 2020, 225, 113483; DOI: doi.org/10.1016/j.enconman.2020.113483

Calculation scope and complexity must be enriched. Simple balance calculations, however their results seem interesting, should be supplemented with detailed parametric studies or optimization.

We, essentially, agree with the reviewer in the sense that this work requires to be continued. Also, we agree with the necessity to perform detailed parametric studies and/or optimization exercises. In fact, we are currently working in different related aspects (development of a specific thermodynamic model, simulation of a reactor model able to reproduce the experimental results, process optimization, economic estimations, etc.). However, it is important to recognize the engineering level of the design proposed to the process in the current work. To the best of our knowledge, no rigorous/extended basic engineering process proposals have been made. The few attempts in literature, which have been cited and discussed in the manuscript, are in the form of case studies, where mass and heat balances and economic estimation are given for particular operating conditions). The main contribution of the current work is to evaluate the process not only at particular operating conditions but in a wide range, thus enabling sensitivity analyses and the determination of the minimum requirements needed to reach energy self-covering.

The process engineering developed in the current work is limited to the first stages of the conceptual design of the process. Consequently, complex and detailed parametric studies and/or optimization are not supported by the current model. The goal of this work is more conceptual and primary as it is reflected in the Abstract. We trayed to solve a primary conceptual question not solved (to the best of our knowledge) up to now, which is finding the potential energy self-covering of the process in wide windows of operating conditions instead of being circumscribed to a particular case study. According to our opinion, the contribution made in this work deserves its publication in a reputed Journal as Applied Sciences. Once we have demonstrated the feasibility of the process from the point of view of energy self-covering, we are planning, for the next future, to propose a more-developed engineering capable for supporting such kind of analysis suggested by the reviewer.

We have improved the instruction section to better describe the results obtained in literature on APR feasibility, both from experimental and simulation works, and the novelty and contribution of the current work.

A wider discussion, encompassing both own results and those presented in relevant related studies should follow the results – see line 83 where comparison of results is promised but this promise is not kept. Comparison of results with other studies is omitted, which is one of the main drawbacks of the manuscript.

We understand the comment of the reviewer, considering the text in line 83 of the original manuscript. It is important to remark that the results of the current work cannot be compared directly to those reported in any scientific paper about APR due to the nature/focus of each work. Most of the works in literature are experimental research studies dealing with catalyst and gas production optimization, processability of feedstocks, reaction mechanism elucidation, thermodynamic considerations, etc. at laboratory scale. This is the case of the works cited in line 83, whereas the current work provides deals with conceptual engineering. To avoid misunderstanding the text in line 83 has been reviewed and rewritten. In addition, the results discussion has been improved by additional comparison with literature originally cited and the new references included in the reviewed version of the manuscript.

 At present the study resembles more an extended computation exercise report than a scientific manuscript. Please do remember that Applied Sciences is a well-established scientific journal in Q2 JCR, publishing relevant research. Therefore, a thorough manuscript revision is needed.

We cordially disagree with the assessment of the current study as an “extended computational exercise report”. Undoubtedly, the work is a computational study, but this feature does not detract from its scientific merits and contribution in the field. The computational research based on the use of process simulation is a well stablished field, with proper methodological rules and forms of expressions. In our opinion we have solved a conceptual problem related to the energetic feasibility of the aqueous phase reforming even when no basic designs of the process are available. It is important to recognize that the experimental research on this process is still mainly at laboratory scale and has been the basis for the operating conditions used in the computational research, as indicated in the manuscript. We have carried out an analysis of the energetic feasibility of the process in potential large-scale developments based solely in a primary conceptual design thanks to process simulation. At this point, it is important to remark that previous works reported in literature have oriented the APR process to H2 production to reach economic sustainability. The current work proposes a plausible alternative for the process, maximizing the route to methane and making energy self-covering feasible, which is particularly relevant in the case of wastewater due to the low organic load of the feedstock. We strongly believe that this conclusion be considered as scientifically valuable.

 Comments and suggestions regarding formatting:

 Please unify the formatting (“10%” or “10 %” as an example). Please do use correct symbols (line 42 and 121 should be ”°C” as an example). Please refer to the equation numbers in the accompanying text.

The units, symbols and other typos have been revised and corrected throughout the manuscript.

 Table 4: see “Methanation conversión”

The word “conversion” has been corrected.

Reviewer 2 Report

Applied Sciences

Toward sustainability of the aqueous phase reforming of wastewater: heat recovery and integration

This study focuses on the assessment of heat integration and sustainability of Aqueous Phase Reforming applied to the treatment of wastewater. In this regard, the authors intended to evaluate the influence of the concentration of organic matter in wastewater and conversion of the reforming and methanation reactions on the overall process using a heat balance.

Please revise the paper within the following points before providing a suitable decision regarding the paper based on MINOR corrections

1.      Section 2.1: please use (sensitive) instead of (sensibility)

2.      Page 5 line 169: please edit this (it affects to the concentration of)

3.      Figure 4: for this sentence (Likewise, the increase in methanation conversion causes a decrease in CO2 concentration in the gas stream). I can see the reverse, CO2 increases with an increase in the mthanation conversion with a decrease in H2 (consumed in the reaction)!

4.      I am not satisfied with the sensitive analysis made in Table 3 regarding the evaluation of wastewater concentration. The authors considered only 1% and 2%. I think it might be feasible to extend this range to present a clearer picture. Also, the pressure of 30 and 50 par might be included

5.      I trust that the addition of statistical results to the Conclusions will be of advantage

Author Response

The authors are grateful for the comments and interest that the reviewer has paid to the manuscript.

This study focuses on the assessment of heat integration and sustainability of Aqueous Phase Reforming applied to the treatment of wastewater. In this regard, the authors intended to evaluate the influence of the concentration of organic matter in wastewater and conversion of the reforming and methanation reactions on the overall process using a heat balance.

Please revise the paper within the following points before providing a suitable decision regarding the paper based on MINOR corrections 

  1. Section 2.1: please use (sensitive) instead of (sensibility)

The term sensibility has been changed by sensitivity.

  1. Page 5 line 169: please edit this (it affects to the concentration of)

The sentence (lines 169-171) has been rewritten as follows: “On the other hand, the conversion values in the reactions involved have influence in the concentration of H2 and alkanes, mainly CH4, in the produced gas, leading to different heating value of the stream.

  1. Figure 4: for this sentence (Likewise, the increase in methanation conversion causes a decrease in CO2 concentration in the gas stream). I can see the reverse, CO2 increases with an increase in the mthanation conversion with a decrease in H2 (consumed in the reaction)!

As the reviewer indicates, there was an error in results discussion in this point. So, this comment has been addressed modifying the manuscript as follows:

“On the contrary, the increase in methanation conversion causes an increase in CO2 concentration in the gas stream, in this case much slighter than observed for H2 concentration.”

  1. I am not satisfied with the sensitive analysis made in Table 3 regarding the evaluation of wastewater concentration. The authors considered only 1% and 2%. I think it might be feasible to extend this range to present a clearer picture. Also, the pressure of 30 and 50 par might be included

The operation conditions showed in Table 3 are not the only ones considered for the sensitive analysis, but the ones showing the best results of this study in terms of potential heat self-covering. Thus, sensitivity analysis for pressure was conducted between 30 and 50 bar, that is the interval in which APR process should be conducted. As indicated above, the description of the objectives of the work and the methodology have been revised to better show the procedures and ranges studied.

Regarding the concentration of organic matter in the wastewater, experimental works (including those carried out in our research group) showed that this concentration must be maintained below 5% in order to maximize the substrate conversion and hydrogen production. The current work complements the experimental work to assessment the feasibility of APR for treating wastewaters. Therefore, concentration values of 1 and 2 % were considered. On the other hand, low concentration feedstocks show the disadvantage of increased heat demand in the reactor due to preheating needs, higher water evaporation, etc. Since the aim of this study is to evaluate the heat balances of the process in, a priori, limiting scenarios, we considered that the use of low concentrations would give more interesting information. In fact, the results in Table 3 show that a concentration of at least 1% is needed to achieve heat self-covering, and that a concentration of 2% could also provide covering of additional demands and losses.

To make this aspect clearer the comments above have been included in the revised manuscript.

  1. I trust that the addition of statistical results to the Conclusions will be of advantage

Following the comments by the reviewer, more quantitative information has been included in the Conclusions.

Round 2

Reviewer 1 Report

Manuscript has improved and most of my recommendations are reflected in the revised version. A few issues need attention in a subsequent minor revision:

- figures 1-3: Try to improve their legibility by increasing text size

- Check Figure 7: replace ","s by "."s as decimal separators

- I am still not convinced regarding the sufficiency of manuscript content (calculation scope and complexity) for its publishing, despite the reasoning of the authors in their answer and the obvious manuscript improvement. I leave the final decision at the disposal of the editor(s).

Author Response

The authors thank the reviewer the comments to improve the manuscript quality.

- figures 1-3: Try to improve their legibility by increasing text size

Figures 1-3 have been increased in size to improve legibility. We considered this option better than increase text size but maintaining the original small size of whole figure.

- Check Figure 7: replace ","s by "."s as decimal separators

Leyend in Figure 7 has been modified to change "," by "."

- I am still not convinced regarding the sufficiency of manuscript content (calculation scope and complexity) for its publishing, despite the reasoning of the authors in their answer and the obvious manuscript improvement. I leave the final decision at the disposal of the editor(s).

Regarding the comment on the complexity and scope of the manuscript, as we commented in the first revision, the process engineering developed focused in the conceptual design of the process. The objective was to assess the potential energy self-covering of the process with a sensitivity analysis covering a range of operating conditions. We consider that the conceptual problem related to the energetic feasibility has been solved, even when no basic designs of the process are available. This is an interesting contribution to the field that can incentive further research on the basis of the demonstrated feasibility. A work with higher complexity, e.g. through detailed parametric studies and/or optimization exercises, would mean to use different methodology and maybe solving some gaps in knowledge not fully supported by data available in experimental works, such us the development of models and the simulation of the catalytic reaction. Such approach would refine the assessment here made, but would be a work different from the one we intended and presented here.